# OpenReview forum: "GeDi: Generative Discriminator Guided Sequence Generation"
_ICLR.cc/2021/Conference — Reject_

### Official Review · AnonReviewer3 · 2020-10-25

**Rating:** 6
**Confidence:** 4

**Review:**

[Summary]
In this paper, the authors propose an efficient method for controllable language generation of large pre-trained LMs (e.g., GPT2). The main idea is to use a smaller, compared to the LM to control, language model trained with control code (Keskar et al., 2019) to generate a per-token score $P(c|x_{1:t}$) to steer the original Language Model distribution. The author proposes two ways, contrastive and discriminative, to approximate $P(c|x_{1:t})$ using bayesian rules. Experiments on open-ended language generation have been shown for positive/negative, detoxification, and topic-control, including a zero-shot topic-control.

[Pros]
- the proposed methodology is novel for the task and it is effective in controlling the desired attributes.
- the proposed method is more efficient than WD (Ghazvininejad et al., 2017) since it does not require a forward to the discriminator for each to token in the vocabulary, and computationally more efficient then PPLM (Dathathri et al. 2020) which requires several updates per token.
- the paper is well-written and easy to follow, except for some minor (later for more info). To the best of my knowledge, the paper is technically correct and reproducible.

[Cons/Question for the authors]
- I have read through the paper, but I could not find any significant test (e.g., annotator agreement, t-test etc.), are the reported human evaluation results significant? could you provide p-values for the results?
- Both detoxification and topic control has no baselines to compare with. For instance, CC-LM-non-tox, CTRL, PPLM could have been used to detoxify the generation. For detox, why not using Universal Triggers (Eric et.al. 2019) for making the model generate toxic text, instead of using prompt from the dev set of the same dataset used for training GEDI?
- Zero-Shot Topics: why PPLM and CTRL cannot do zero-shot on a topic (from the conclusion)? PPLM can use a bag-of-word discriminator, so no training required and generate any kind of topics, and CTRL can use different link/prompt to generate unseen topics?
- Greedy decoding: why using greedy decoding for a language generation task? it is well known that top-p and top-k greatly improve the model generation, are there performance drop if using top-p? how GEDI compare to CTRL, PPLM  in this setting?

[Reason to accept]
The proposed method is a simple and effective way to control the generation of large language models. This is an important and timely problem, especially for language detoxification.

[Reason to reject]
The experiments are a bit unclear, looking forward to the author response

[Suggestions and some more questions]
- With reference to the sentence: " In addition to class-conditional generation, CC-LMs can be used as generative classifiers by applying Bayes rule to compute $P(c|x_{1:T})$, as is done by Keskar et al. (2019) for source attribution." Could you please add the inline formula, $p_θ(c|x) \approx p_θ(x|c)p(c)$, it saves one jump to the paper and makes the paper more readable :)
- Could you please elaborate on why GEDI would be 10k fold less computation as compared with a unidirectional classifier? Could you include a more detailed computational cost analysis?

---

> ### Author Response · Authors · 2020-11-24
> **Response to Reviewer 3**
>
> Thank you for your review!
>
> *"I have read through the paper, but I could not find any significant test (e.g., annotator agreement, t-test etc.), are the reported human evaluation results significant? could you provide p-values for the results?"*
>
> All of the larger improvements in the previous version of the paper were statistically significant, but since we rerun them we’ve added significance tests for the paper in Appendix E. We used a Wilcoxon signed rank test for statistical significance to compare matched pairs since all models generate from the same set of prompts (and because a non-parametric test is appropriate for ordinal data).
>
> *“Both detoxification and topic control has no baselines to compare with. For instance, CC-LM-non-tox, CTRL, PPLM could have been used to detoxify the generation. For detox, why not using Universal Triggers (Eric et.al. 2019) for making the model generate toxic text, instead of using prompt from the dev set of the same dataset used for training GEDI?”*
>
> Thanks for the suggestions. To address these concerns about baselines and triggers, we added a new detoxification experiment that uses triggers from Real Toxicity Prompts and uses PPLM and CTRL-style baselines. We used triggers from Real Toxicity Prompts that were non-toxic but resulted in a high probability of GPT-2 generating toxic text. We found that for detoxification, GeDi performed on par with PPLM for greedy decoding (with repetition penalty for both), and performed statistically significantly better than PPLM with top-p sampling (while being 30x faster). We also found GeDi could detoxify generation from GPT-3 in this new experiment (PPLM can't be applied to GPT-3 with OpenAI's API, but GeDi can).
>
> *“Zero-Shot Topics: why PPLM and CTRL cannot do zero-shot on a topic (from the conclusion)? PPLM can use a bag-of-word discriminator, so no training required and generate any kind of topics, and CTRL can use different link/prompt to generate unseen topics?”*
> We took out our more general claims and focus specifically on comparing with class conditional LMs. For PPLM, bag of words list still needs to be specified somehow and likely would not be available for any imaginable topic. But presumably, it might be possible to choose words that are close to the desired zero-shot topic in embedding space. For CTRL, we attempted to use zero shot control codes using prompting words. It’s possible that links could work better, but one would need to specify a full link (which could influence the generation in other ways) and couldn’t simply use a word.
>
> *"Could you please add the inline formula, pθ(c|x)≈pθ(x|c)p(c), it saves one jump to the paper and makes the paper more readable :)”*
>
> Good idea, :) We added this.
>
> *“Greedy decoding: why using greedy decoding for a language generation task? it is well known that top-p and top-k greatly improve the model generation, are there performance drop if using top-p? how GEDI compare to CTRL, PPLM in this setting?”*
>
>
> Greedy decoding on its own tends to lead to degenerate text, but Keskar et al. (2019) suggests  that greedy decoding works well with a repetition penalty. To examine the effect of sampling, we decided to include results with top-p sampling with p=0.9 for GPT-2, PPLM, and GeDi, and found that GeDi could work well in that setting too. The samples in our new experiments resulting from greedy decoding with a repetition penalty were rated in human annotations as having a higher fluency than the top-p samples for all models where we compared both settings.
>
> *"Could you please elaborate on why GEDI would be 10k fold less computation as compared with a unidirectional classifier? Could you include a more detailed computational cost analysis?"*
>
> We've added pseudo code for GeDi in Section 3 that might help make it more clear how GeDi can do this efficiently. The objective is to classify the resulting sequence that would occur for every possible next token for a partial sequence. To compute this with a standard classifier, every token in the vocabulary would need to be passed through the classifier as an input, each in a separate forward pass. For a vocab size of 50k, this would require 50k forward passes. However, GeDi can compute these classification probabilities for every possible next token via Bayes rule ($p(c|x) \propto p(x|c)p(c)$ as you mentioned) using only element wise operations in the output layer. This requires only 2 forward passes through the GeDi, one for each of the two control codes.

---

### Official Review · AnonReviewer2 · 2020-10-26
**A novel approach but limited experiments with a lack of valid experimental setups.**

**Rating:** 5
**Confidence:** 4

**Review:**

##########################################################################
Summary:
This paper proposed using small-sized LM as a generative discriminator to guide large-sized LM for better controllability and decoding efficiency.

##########################################################################
Reasons for score:

My score is marginally below the acceptance threshold.

Pros:

1. The most exciting part of this paper is to factor out the opposite labels of each token (e.g., positive and negative, or toxic or non-toxic) using Bayes rule in generative models.


Cons:

1. The contribution of controllability and producing safe output should be separated out. Safer LMs seem to be the outcome of the controllability of the LM by canceling out the opposite part of the target label using the Bayes rule. Highlighting this point might be helpful for showing out the novelty and value of this work. The current frame of the work seems quite distributed between applications and architectural contributions.

2. The main concern of this work is the lack of focused contributions and their validations. The authors claim that this model is good at almost everything; efficiency, controllability while maintaining linguistic quality, reducing the toxicity of GPT2, zero-shot topical generation, etc. However, in fact, most of the experiments in Section 5 are very shallow, uncontrolled, and lack statistical significance. I appreciate the general effectiveness of the model and I don’t doubt it. However, as a conference paper with limited pages, it would be better to make one or two points among them and providing more in-depth with valid setups of experiments. I put additional notes about the experiments below.

3. As mentioned above, the novelty of this comes from using the Bayes rule to make positive and negative labels far from each other. The authors also mention that this is a sort of contrastive learning, and they used contrastive generation. However, contrastive learning is often used to refer to learning by separating two different instances out far each other. In this work, there is no such auxiliary optimization during training time, but the posteriors are re-weighted using the Bayes rule. I would recommend using a more exact term to describe this rather than contrastive generation.

4. In Section 3.1.1, various heuristics are used. I expected to see the effect or ablation of each heuristic and how important each of them is in terms of generation quality. Also, the baseline models such as CTRL, CC-LM, and PPLM seem to be not using the same heuristics, which seems to be not fair.

5. The output in Table 6 makes me doubt how the experiments are badly controlled. The outputs from positive and negative sentiment are totally different and almost random text, meaning that the content of the generators is not controlled properly. In preparation for prompts for GeDis (5.2) or for measurement of label fidelity (5.3), authors used the pre-trained BERT or RoBERTa on the target attribute like toxicity and topics. As far as I know, these automatic classifiers are not correlated with human judgment, in fact, leading to huge wrongly-predicted labels. I wonder why human annotations are not used here.

6. In Table 2, I don’t see any significant improvements of GeDi against PPLM in its attribution score (i.e., positivity) and transferability to the target domain. Similar to the comment above, none of the experiments are controlled in content. Measuring how the text is similar to the domain (e.g., book-like) sounds interesting but there are no further details of how the human evaluation is studied, what kinds of guidelines are provided to annotators, how the output looks like, etc.

7. In Table 3 and 4, have you performed the same experiments with PPLM and CTRL?

##########################################################################

---

> ### Author Response · Authors · 2020-11-24
> **Response to Reviewer 2**
>
> Thanks for your review. Our update to the paper was largely aimed at improving our experimental set up, and we now include baselines, significance tests, and ablation studies, as we will describe in our replies below and also in our general comment that gave an overview of our update.
>
> *"The contribution of controllability and producing safe output should be separated out. Safer LMs seem to be the outcome of the controllability of the LM by canceling out the opposite part of the target label using the Bayes rule. Highlighting this point might be helpful for showing out the novelty and value of this work. The current frame of the work seems quite distributed between applications and architectural contributions."*
>
> It is indeed true that safer LMs are the outcome of the controllability, and that this is a consequence of using Bayes rule to cancel out predictions resulting from opposing control codes. We chose to also emphasize LM safety since it is such a big concern right now, and we think this application will be valuable.
>
>
> *"I would recommend using a more exact term to describe this rather than contrastive generation“*
>
> We have removed this term
>
>
> *"In Section 3.1.1, various heuristics are used. I expected to see the effect or ablation of each heuristic and how important each of them is in terms of generation quality. Also, the baseline models such as CTRL, CC-LM, and PPLM seem to be not using the same heuristics, which seems to be not fair.”*
>
> We‘ve added ablation studies for the weighted decoding and filtering heuristic we use. PPLM also had many heuristics in its design, including KL-scaling, post-geometric norm-fusion, early stopping of latent updates, and adaptive gradient normalization. Also note that GeDi in its current simplified form only has 2 hyper-parameters, whereas PPLM has 7. Also, these heuristics would not be applicable to CC-LM or CTRL, we are using heuristics specific to weighted decoding schemes.
>
>
> *“The output in Table 6 makes me doubt how the experiments are badly controlled. The outputs from positive and negative sentiment are totally different and almost random text, meaning that the content of the generators is not controlled properly.”*
>
> In unconstrained text generation from language models like GPT-2 with short prompts, it is normal for generations to be very different even under slightly different settings. In this case, we would definitely expect the positive and negative to be very different as we are guiding towards different attributes.
>
>
> *“In preparation for prompts for GeDis (5.2) or for measurement of label fidelity (5.3), authors used the pre-trained BERT or RoBERTa on the target attribute like toxicity and topics. As far as I know, these automatic classifiers are not correlated with human judgment, in fact, leading to huge wrongly-predicted labels. I wonder why human annotations are not used here.”*
>
> Unfortunately it is impossible to run human evaluation experiments for every possible design decision and hyper-parameter setting. We did find across tasks that classification labels given by RoBERTa were well but not perfectly correlated with human evaluation annotations.
>
> *"In Table 2, I don’t see any significant improvements of GeDi against PPLM in its attribution score (i.e., positivity) and transferability to the target domain. Similar to the comment above, none of the experiments are controlled in content. Measuring how the text is similar to the domain (e.g., book-like) sounds interesting but there are no further details of how the human evaluation is studied, what kinds of guidelines are provided to annotators, how the output looks like, etc."*
>
> We now provide statistical significance p-values in Appendix E .  GeDi achieves statistically significantly better sentiment control than PPLM for top-p sampling (both negative and positive), and statistically significantly better negative sentiment control than PPLM in the greedy setting, while also achieving generation speeds 30 times faster. We also now provide the exact annotator guidelines in Appendix G.
>
> *“In Table 3 and 4, have you performed the same experiments with PPLM and CTRL?”*
>
> We’ve added PPLM and CTRL-style baselines to our detoxification experiments, and CTRL style baselines to our topic experiments.

---

### Official Review · AnonReviewer4 · 2020-10-28
**Review: GeDi: Generative Discriminator Guided Sequence Generation**

**Rating:** 4
**Confidence:** 5

**Review:**

#### Summary

The authors propose a method for controlling attributes of generated text (sentiment, topic, toxicity, etc) by reweighting a base language model's token-level distributions with auxiliary conditional language models, bayes rule, and additional heuristics.

The core idea is simple - which is a strength in my view - and does not require retraining the base language model, which could be important as language models become more expensive to train. However, the clarity and experiments in this paper fall short: the experimental setup has issues (detailed below), the effect on perplexity is quite large but relegated to the Appendix, several claims are speculative and lacking corresponding experimental evidence, and it is unclear how the additional heuristics affect performance.

The method seems promising, but with the current experiments it is difficult to draw conclusions about how the method affects performance and which parts of it are necessary; given that this is an empirical paper, I would therefore not recommend acceptance in its current form.

#### Experimental setup

- **Human evaluation**. There is little information given about how the human evaluation prompts are written (and why they are written that way), how many evaluators are used, and there are no significance tests (e.g. "We run human evaluation to measure toxicity" does not given enough details). This is concerning since all of the results in the main text use human evaluation, sometimes with small differences between methods.

- **Automatic eval/perplexity**. The authors only measure perplexity in one set of experiments in the appendix (and it gets worse by introducing GeDI training). It would be good to have perplexity and automatic metrics to compare against the human evaluation (e.g. see Table 4 in the PPLM paper).

- **Decoding algorithms**. The authors only show results for greedy decoding with a repetition penalty (with no ablation on the choice of penalty parameter). Results with a sampling method (e.g. nucleus) are needed for this open-ended setting, or an argument for why these aren't considered.

#### Effect of the method
- **Effect on perplexity**. The perplexity gets much worse as the gedi training is introduced (i.e. $\lambda$ decreases), e.g. going from 25 to 45 on IMDb. This result is in the Appendix, and perplexity is never evaluated/reported in the other experiments.

- **Gedi training**. It's unclear whether the gedi training (i.e. the $\mathcal{L}_d$ loss) is beneficial: in some experiments $\lambda=1.0$ performs similarly, and on the IMDb/MNLI/QNLI experiments decreasing $\lambda$ either hurts, has no effect, or improves performance (i.e. no consistent trend).

- **Detoxifying**. It's unclear how significant the results in Table 3 are, and there are no baselines; in general it it difficult to draw conclusions from these results.

#### Speculative claims or conclusions
- **Domain transfer**. Figure 1 gives an intuition for why domain transfer might be possible, but only an anecdote (first paragraph of 5.1, "we noticed that") and a single experiment is done, where the method performs similarly to PPLM. Crucially, the GEDI training does not appear to help over just re-weighting with the conditional LM ($\lambda=1.0$ vs. $\lambda=0.6$). Could the authors comment on this result? How well does domain transfer work for less similar domains? How is perplexity affected for the models reported in Table 2?

- **Zero-shot control codes**. The authors only provide anecdotes for evaluating the Zero-shot control codes. Based on the evaluation it's quite speculative to say "GeDi’s ability to generalize to new control codes zero-shot gives the ability to generate text corresponding to many topics and subtopics.".

- **Smaller language models guiding larger language models**. To be fair, the authors use GPT-2 medium as the conditional language model, and GPT-2 XL as the base language model, which is larger, but there was no investigation of this aspect of size difference. How small can the conditional LM be? Why was medium used instead of small? What if large was used? Does the conditional LM need to be a large-scale pretrained model (it would be nice to see a baseline of a simpler conditional LM)?

#### Heuristics
- Several heuristics are used: $\alpha/T_i$ weighting, $\omega$ weighting, nucleus filtering, keeping tokens over a threshold, repetition penalty, and rescaling the logits to positive (used in only one experiment).
- How does each of these affect performance? There are no ablations, and given the small differences in some of the experiments it is unclear whether performance would actually be worse if we changed one of the heuristics. One outcome may be that the method only works for a careful balance of hyperparameters, which could be fine, but we don't have a sense of the variation.

---

> ### Author Response · Authors · 2020-11-24
> **Response to Reviewer 4**
>
> Thanks for your in-depth feedback on our work!
>
> **re: "Human evaluation”**
>
> For our latest experiments, we’ve added p-values for all important significance tests to Appendix E  and screen shots of annotator instructions to Appendix G.
>
> **re: "Automatic eval/perplexity"**
>
> Those perplexity numbers in the appendix of the original submission (now removed because we no longer consider GeDi training) are actually measuring perplexities of real data under the discriminator LM, whereas table PPLM measured the perplexities of their model’s samples under GPT-2. We’ve added perplexity scores and automatic metrics to Tables 2 and 3 to compare with human evaluations as per your suggestion.
>
> **re: “Decoding algorithms"**
>
> We’ve now added results for the most critical models with top-p sampling, and find that GeDi works well in this setting. Our human evaluation experiments also found higher fluency scores for our original  generation method all models that compared both settings, so we still focus primarily on that setting.
>
> **re: “Detoxifying”**
>
> In the previous toxicity experiments the reductions to GPT-2 toxicity were strongly statistically significant. We’ve run new detoxification experiments to compare many more settings with more baselines, with significance results in appendix G.
>
> **re: “Domain transfer"**.
>
> Our main point about domain transfer is that CTRL-style models (or any model that finetunes to new data) will only be able to generate text that resembles text from the training domain. So a CTRL-style model trained for sentiment on movie reviews will only be able to control sentiment on movie reviews, which we show by the high percentage of movie reviews generated by the model even when given unrelated prompts. This significantly reduces the utility of CTRL-style models. These experiments show that Discriminator based methods like GeDi and PPLM do not appear to have this problem. We may not be able to generalize sentiment to every possible domain, but the point is that the model rarely if ever reverts back to the domain where the discriminator was trained on, allowing more of the breadth of the original LM (in this case GPT-2) to be retained.
>
> **re: "Zero-shot control codes"**
>
> We added evaluations in Section 5.3 where we train the GeDi on only 3 out of 4 AG news classes and hold one out (We do this separately for all 4 classes). We then evaluate generations on the held out zero-shot control codes by measuring how often a classifier (trained on all 4 classes) classifies them as the same class as the control code, and show that zero-shot control codes are able to bias generation towards the desired class.
>
> **re: “Smaller language models guiding larger language models."**
>
> Our main point here is about the size of the base-LM rather than the size of the GeDi, since the base-LM can potentially be huge. Finetuning large language models is very expensive, and it is much more convenient to be able to finetune a smaller model and use it to control a large LM. To better illustrate the effectiveness of this, we included results using our GPT-2-medium based GeDi  (345M parameter) to detoxify GPT-3 (175B parameter).
>
>
> **re: "Heuristics"**
>
> We’ve simplified our method as it was more complicated than it needed to be before. It now only has two hyper-parameters (in comparison, PPLM has 7), since we removed GeDi training ($\lambda$), decided to omit the bias parameter (which was only used with a GeDi trained model in the original submission), and decided to remove the heuristic that protects tokens from filtering ($\tau$). We’ve also added an ablation study (Appendix D) comparing applying our filtering heuristic, our weighted decoding heuristic, and our combined filtering weighted decoding heuristic. Our results show that combining these heuristics likely isn’t critical but may be slightly helpful in some settings.

---

### Official Review · AnonReviewer1 · 2020-10-29
**The paper is written well**

**Rating:** 6
**Confidence:** 4

**Review:**

The paper proposed a method —- GeDi — to generate guided and controlled text from a large language model (LM). The method utilizes smaller LMs as generative discriminators to guide generation from large LMs to make them safer and more controllable. By safer and controllable they emphasis on the toxicity, hate, bias, and negativity contains in the training of the large LM. The proposed method guides generation at each time step by computing classification probabilities for all possible next tokens via Bayes rule by normalizing over two class-conditional distributions (i.e. contrastive discrimination); one conditioned on the desired attribute, or control code, and another conditioned on the undesired attribute (i.e. contrastive attribute), or anti-control code.

The paper explores ways to increase generation speed and claimed that with the proposed techniques the generation speeds more than 30 times faster compared to PPLM model. The paper explores different heuristics to impose the guided generation including bias parameter, weighted decoding and filtering heuristics. The findings are that GeDi gives stronger controllability than the state of the art method (i.e.PPLM, CC-LM, CTRL).

Experiments show that, training GeDi on four topics (i.e. Business, Science/Tech, Sports, World ) allows the controlled generation of new topics zero-shot from just a keyword. They also demonstrate that GeDi can make GPT-2 (1.5B parameters) significantly less toxic without sacrificing linguistic quality.


Re: “so long as the LM and GeDi share the same tokenization”: can you please elaborate the constraint on ‘same tokenization’?

Re: “If the GeDi was trained on movie reviews for sentiment control, its direct class-conditional predictions will be biased towards predicting movie review words (illustrated by next word prediction of “cinematic”). However, by contrasting the predictions of opposing control codes via Bayes rule, the bias towards movie reviews can be cancelled out.”: The word cinematic can reveal a neutral/negative sentiment, is there any possibility that pushing the sentiment towards positive might degrade the accuracy of the overall generation?

Re: GeDi training (λ < 1 in Equation (10)) and standard generative training(λ = 1 in Equation (10)). : How the value for λ = 0.6 was chosen? What is the impact of other values for this hyper-parameter?

Re: “In order to have prompts that are more likely to trigger aggressive generations but less likely to be explicitly toxic, we pass candidate prompts through a RoBERTa (Liu et al., 2019) model trained to classify toxicity, and only kept prompts where RoBERTa was less confident about the toxicity label.“: how did you measure model confidence about the toxicity label?

---

> ### Author Response · Authors · 2020-11-24
> **Response to Reviewer 1**
>
> Thank you for your review and your questions!
>
> *“can you please elaborate the constraint on ‘same tokenization’?”*
>
> The GeDi and the base language model that the GeDi guides must share the same vocabulary in order to apply Bayes rule to compute the attribute classification probabilities for every candidate next word. If the output vocabularies were different, then it wouldn’t be possible to use the classification probabilities given by GeDi to re-weight the language models predictions. Fortunately for us, all GPT-2 models (as well as GPT-3, which we use in some of our new experiments) use the same vocabulary.
>
> *“is there any possibility that pushing the sentiment towards positive might degrade the accuracy of the overall generation?”*
>
> In controllable generation methods like GeDi and PPLM, there is a tradeoff between how aggressively you steer generation, and the quality of the text that is generated (where the aggressiveness of the steering is controlled by hyperparameters). So over aggressively steering generation can negatively affect generation quality. We set hyper-parameters to maintain generation quality while also steering towards the desired attribute.
>
> *“How the value for λ = 0.6 was chosen? What is the impact of other values for this hyper-parameter?”*
>
> We decided to remove GeDi-training from the paper to simplify the method (so all of the new experiments are equivalent to  λ = 1. In the previous version, lower values of lambda used a stronger discriminative loss, and we found  λ =0.6 and 0.8 to give the most reliable attribute control in the GeDi training settings that we tested. We chose 0.6 because it was more different from generative training.
>
> *"how did you measure model confidence about the toxicity label"*
>
> We used the confidence of the RoBERTa model in terms of its output probabilities and took the prompts that it was least sure about the output class. Our toxicity experiments have changed though, and we have a new procedure for selecting prompts. We use prompts from RealToxicityPrompts that are classified as non-toxic, but have a high probability of resulting in toxic generation from GPT-2. We changed this due to another reviewer concern that the prompts came from the dev set of the same data we used to train the GeDi. We still found that GeDi was very effective for detoxifying GPT-2 (and also GPT-3) in our new experiments in this setting.

---

### Official Review · AnonReviewer5 · 2020-11-07
**Proposes algorithm for controllable sequence generation**

**Rating:** 5
**Confidence:** 3

**Review:**

Summary
The paper considers the problem of attribute-based sequence generation, particularly in language models. Authors propose a framework “GeDi” which learns a generative classifier for controlling generation from a large language model. With experiments on publicly available datasets and models, and including human-evaluation, the authors empirically demonstrate that the algorithm is computationally efficient and is competitive against strong baseline algorithms like CTRL, Plug&Play language models (PPLM).

Reason for the score
I vote for rejecting the current version of the paper (marginally below acceptance threshold). While the premise of the problem is well motivated, I think several sections of the paper are difficult to follow. I would strongly encourage the authors to include a pseudo code of the algorithm to improve the presentation of the central idea. The paper includes several experiments, though I think some critical ablations are missing.

Strengths
+ The problem is practically well motivated and is very relevant to the language learning community.
+ The proposed algorithm of using generative classifiers is computationally efficient compared to strong baselines like CTRL and PPLM. The experimental results suggest that the algorithm also allows for better control over generation from a LM while maintaining linguistic quality.

Weaknesses
- Several sections of the paper are hard to follow. To improve the presentation of the idea, I would encourage the authors to distill the central idea into a pseudo code which goes along with Section 3.
- The experiments on detoxification are critical to the thesis of the paper, however it seems that experiments in Section 5.2 consider only GPT-2 baselines? I think a strong baseline based on prior-work, like a CTRL generator conditioned on the positive label (as mentioned in Introduction), would help evaluating the gap between proposed approach and current algorithms.

---

> ### Author Response · Authors · 2020-11-24
> **Response to Reviewer 5**
>
> Thanks for your review! Here are responses to your main concerns:
>
> *“I think some critical ablations are missing.”*
>
> We have simplified some aspects of our method to make it easier to study the components. We removed “GeDi training” as it complicates the method and was not critical to making the main idea of the paper (GeDi-guided generation) work. We then provided an ablation study of the methods we do use in Appendix D.
>
>
> *“To improve the presentation of the idea, I would encourage the authors to distill the central idea into a pseudo code which goes along with Section 3”*
>
> Thanks for the suggestion. We’ve added pseudo code in Section 3.
>
> *“The experiments on detoxification are critical to the thesis of the paper, however it seems that experiments in Section 5.2 consider only GPT-2 baselines? I think a strong baseline based on prior-work, like a CTRL generator conditioned on the positive label (as mentioned in Introduction)”*
>
> We decided to run new more extensive detoxification experiments to better support the thesis of our paper. Our new toxicity experiments use prompts from Real Toxicity Prompts. We include a PPLM (which is still 30x slower for generation than GeDi in this setting) and a CTRL-style (CC-LM) baseline conditioned on the non-toxic label. We find in these experiments that the CTRL-style baseline, while somewhat effective for detoxification, significantly reduces the fluency of generation in human evaluation, which is likely a result of finetuning the CC-LM to new data (which also severely limits the diversity of the model since the toxicity dataset we use is composed mostly of Wikipedia comments). We also included some results detoxifying GPT-3, where existing baselines such as CTRL-style or PPLM cannot be applied through Open AI’s API (but GeDi can).

---

### Public Comment · ~Hyunwoo_Kim3 · 2020-11-16
**The Rational Speech Acts framework**

Hello, I read the paper very interestingly!

The idea of controlling language models by computing the posterior distribution with a generative discriminator is very widely studied as the Rational Speech Acts (RSA) framework [1] in computational pragmatics.

The RSA framework coins the generative discriminator as a listener and the language model as the speaker.
It treats language use as reasoning between probabilistic speaker and listener.
This idea has been applied to various tasks in NLP, including text-generation [2], reference games [3], image captioning [4-6], instruction following [7], translation [8], referring expression generation [9], and dialogue.

It would be really great if the authors align the paper with this long line of work in computational pragmatics!



* [1] Michael C Frank and Noah D Goodman. 2012. Predicting Pragmatic Reasoning in Language Games. Science, 336(6084):998–998.
* [2] Sheng Shen, Daniel Fried, Jacob Andreas, and Dan Klein. 2019. Pragmatically Informative Text Generation. In NAACL-HLT.
* [3] Jacob Andreas and Dan Klein. 2016. Reasoning about Pragmatics with Neural Listeners and Speakers. In EMNLP.
* [4] Junhua Mao, Jonathan Huang, Alexander Toshev, Oana Camburu, Alan L Yuille, and Kevin Murphy. 2016. Generation and Comprehension of Unambiguous Object Descriptions. In CVPR.
* [5] Ramakrishna Vedantam, Samy Bengio, Kevin Murphy, Devi Parikh, and Gal Chechik. 2017. Context-Aware Captions from Context-Agnostic Supervision. In CVPR.
* [6] Reuben Cohn-Gordon, Noah Goodman, and Christopher Potts. 2018. Pragmatically Informative Image Captioning With Character-level Inference. In NAACL-HLT.
* [7] Daniel Fried, Jacob Andreas, and Dan Klein. 2017. Unified Pragmatic Models for Generating and Following Instructions. In NAACL-HLT.
* [8] Reuben Cohn-Gordon and Noah Goodman. 2019. Lost in Machine Translation: A Method to Reduce Mean- ing Loss. In NAACL-HLT.
* [9] Sina Zarrieß and David Schlangen. 2019. Know What You Don’t Know: Modeling a Pragmatic Speaker that Refers to Objects of Unknown Categories. In ACL.

---

> ### Author Response · Authors · 2020-11-24
> **Reply to RSA related work**
>
> Thanks for pointing out these papers. The RSA framework idea of an interactive game between a listener and speaker does conceptually relate to our work, and uses Bayes rule in similar way.  It seems that the biggest difference between GeDi and previous RSA based approaches is that GeDi use a separate discriminator (that acts as a listener) trained to isolate a specific attribute, and generalizes this attribute by guiding a more general base language model. We now mention several of these papers in the related work.

---

### Author Response · Authors · 2020-11-24
**Overview of changes**

For convenience, here is an overview of the main changes we have made to the paper:

* We removed GeDi training from the paper because we felt it subtracted from the core idea of GeDi guided generation. GeDi training results in GeDis that are stronger classifiers, but it complicates the method and the actual advantages in controllability are very minor at best, so we decided to exclude it and focus on GeDi-guided generation with generatively trained models. This also makes it easier to do ablation studies, which several reviewers asked for.

* We include new detoxification experiments (Section 5.2) using triggers from RealToxicityPrompts [1] that lead to high toxicity in GPT-2. We include PPLM and CC-LM (CTRL-style) as baselines. Overall, GeDi was effective (very strongly statistically significant) for detoxifying GPT-2, and performed similarly to PPLM in the greedy setting and better than PPLM in the top-p setting (while also achieving generation speeds 30 times faster). To emphasize our point about efficiently controlling very large language models, we also show that we can use a 345M parameter GeDi to greatly reduce toxicity in generations 175B parameter GPT-3, which we accomplish by using GeDi to modify decoding using OpenAI’s GPT-3 API.

* We include ablation studies of our two main heuristics using automatic metrics in Appendix D.

* We include pseudocode to describe the method in Section 3.

* We include statistical significance tests for human evaluation experiments in Appendix E.

* We give screen shots of exact annotator instructions in Appendix G.

* We include top-p sampling in our sentiment and detoxification experiments for our GPT-2, GeDi, and PPLM models. Our main findings were 1. Our original generation method (greedy with a repetition penalty) results in higher fluency scores across models and tasks. 2. GeDi can still control generation well compared with baselines when using top-p sampling.

* We include new topic experiments in Section 5.3 to quantitatively show (using automatic metrics) that zero-shot control codes can bias generation towards unseen topics. We do this by training GeDi models with a class held out from the training set, and showing that we can bias generation towards held out classes by conditioning on an unseen control code.

[1] Gehman, S., Gururangan, S., Sap, M., Choi, Y., & Smith, N. A. (2020). Realtoxicityprompts: Evaluating neural toxic degeneration in language models. EMNLP 2020.

---

### Decision · Program_Chairs · 2021-01-07
**Final Decision**

**Decision:**

Reject

**Comment:**

Reviewer #2 has written a nice summary of the paper which I quote below.

“The core idea is simple - which is a strength in my view - and does not require retraining the base language model, which could be important as language models become more expensive to train. However, the clarity and experiments in this paper fall short: the experimental setup has issues, the effect on perplexity is quite large but relegated to the Appendix, several claims are speculative and lacking corresponding experimental evidence, and it is unclear how the additional heuristics affect performance.

The method seems promising, but with the current experiments it is difficult to draw conclusions about how the method affects performance and which parts of it are necessary; given that this is an empirical paper, I would therefore not recommend acceptance in its current form.”

Key Strengths
+ Well-motivated problem of considerable interest
+ A relatively straightforward Bayesian solution
+ Proposed solution is computationally efficient compared to other competing approaches.

The paper has been thoroughly reviewed by the reviewers and as a result numerous questions has surfaced. While the authors addressed most of the questions adequately, there are still many unanswered questions. They include:
- Readability issues highlighted by Reviewer #1
- Reviewer #1: “"how did you measure model confidence about the toxicity label"
- Reviewer #4: The perplexity gets much worse as the gedi training is introduced (i.e. Λ decreases), e.g. going from 25 to 45 on IMDb. This result is in the Appendix, and perplexity is never evaluated/reported in the other experiments.
- Reviewer #4: Crucially, the GEDI training does not appear to help over just re-weighting with the conditional LM ( vs. ). Could the authors comment on this result? How well does domain transfer work for less similar domains? How is perplexity affected for the models reported in Table 2?
- Reviewer #4: How small can the conditional LM be? Why was medium used instead of small? What if large was used? Does the conditional LM need to be a large-scale pretrained model (it would be nice to see a baseline of a simpler conditional LM)?
Several heuristics are used:  weighting, nucleus filtering, keeping tokens over a threshold, repetition penalty, and rescaling the logits to positive (used in only one experiment).
- How does each of these affect performance? There are no ablations, and given the small differences in some of the experiments it is unclear whether performance would actually be worse if we changed one of the heuristics. One outcome may be that the method only works for a careful balance of hyperparameters, which could be fine, but we don't have a sense of the variation.
- Reviewer #2: The output in Table 6 makes me doubt how the experiments are badly controlled. The outputs from positive and negative sentiment are totally different and almost random text, meaning that the content of the generators is not controlled properly.